# Comparative Mortality and Adaptation of a Smurf Assay in Two Species of Tenebrionid Beetles Exposed to *Bacillus thuringiensis*

**DOI:** 10.3390/insects11040261

**Published:** 2020-04-24

**Authors:** Caroline Zanchi, Ana Sofia Lindeza, Joachim Kurtz

**Affiliations:** Institute for Evolution and Biodiversity, University of Münster, Hüfferstraße 1, 48149 Münster, Germany

**Keywords:** *Bacillus thuringiensis tenebrionis*, *Bacillus thuringiensis tolworthi*, *Tenebrio molitor*, *Tribolium castaneum*, Smurf assay, oral infection

## Abstract

*Bacillus thuringiensis* is a spore-forming bacterium which infects insect larvae naturally via the oral route. Its virulence factors interact with the epithelium of the digestive tract of insect larvae, disrupting its function and eventually leading to the death of susceptible hosts. The most cited *B. thuringiensis* killing mechanism is the extensive damage caused to the insect midgut, leading to its leakage. The mortality caused by *B. thuringiensis* has been shown to vary between serovars and isolates, as well as between host life stages. Moreover, whether susceptibility to *B. thuringiensis*-induced gut leakage is generalized to all host species and whether there is individual variation within species is unclear. In this study, we adapted a non-invasive “Smurf” assay from *Drosophila melanogaster* to two species of tenebrionid beetles: The mealworm beetle *Tenebrio molitor* and the red flour beetle *Tribolium castaneum*, during exposure to *B. thuringiensis*. We highlight a differential mortality between two age/size classes of *T. molitor* larvae, as well as different killing dynamics between *B. thuringiensis* var. *tenebrionis* and var. *tolworthi* in *T. castaneum*. The Smurf assay did not reveal a high occurrence of extensive gut disintegration in both host species upon ingestion during *B. thuringiensis* exposure.

## 1. Introduction

*Bacillus thuringiensis* is the most used bioinsecticide to date. This sporulating bacterium expresses a variety of virulence factors showing insecticidal activity at several stages of its growth, such as cytolytic, proteolytic and chitinolytic enzymes, which facilitate the degradation of the peritrophic matrix and midgut epithelial cells [1]. The best studied virulence factor is a crystalline inclusion composed of monomeric protoxins (*Cry* and *Cyt* toxins), which upon ingestion by the insect are activated by gut proteases. The resulting delta-endotoxins then bind specific receptors on the brush border membrane vesicles of midgut epithelial cells, eventually causing their lysis. The binding affinities to host receptors as well as the synergy between them confers each strain with a certain degree of host specificity [2,3]. After host death, *B. thuringiensis* cells switch to a saprophytic lifestyle and exploit the host cadaver until they sporulate, releasing infectious spores and crystals into the environment [4].

The most frequently cited killing mechanism of *B. thuringiensis* is the death by septicemia, where at a relatively early time point in the infection process, the epithelium of the gut of the insect host is disintegrated to an extent that allows the migration of gut contents, fluids and/or bacteria to the body cavity (i.e., the hemocoel). It is the subsequent establishment of the infection by these bacteria in the hemocoel that causes host death [1,3,5,6]. The migration of vegetative *B. thuringiensis* cells from the midgut to the hemolymph indeed seems to be a crucial aspect of the infection. In the nematode *C. elegans*, for example, the damage caused by the virulence factors of *B. thuringiensis*, in particular the delta-endotoxin Cry5Ba, to the midgut epithelium, results in an extensive invasion of the body cavity of the worm, whereas avirulent strains stay confined to its midgut [7,8].

However, the timing of the induced gut leakage relative to host death seems to vary between host-pathogen systems. In the European corn borer *Ostrinia nubilalis*, the vegetative cells of *B. thuringiensis* “pile up” on the basement membrane of the midgut epithelium in live larvae, but are found in the hemolymph of newly dead ones, indicating that the invasion of the hemocoel indeed caused host death [9]. In *Spodoptera littoralis*, lesions caused by the delta-endotoxin Cry1Ca allow the invasion of the hemocoel of live hosts by gut bacterial cells which then proliferate and kill through septicemia [10]. In *Plutella xylostella*, exposing larvae to spores and delta-endotoxins results in the presence of spores in the hemolymph [11]. By contrast, in the rice meal moth larva *Corcyra cephalonica*, germination of spores and proliferation of vegetative cells is confined to the midgut and does not cross the epithelium surface before host death [12]. Similarly, when the Coleoptera-specific strain *B. thuringiensis tenebrionis* was fed to larvae of the cottonwood leaf beetle *Chrysomela scripta*, the authors could observe a destruction of the midgut epithelial cells and a proliferation of vegetative cells in the gut lumen, but did not observe a septicemia in the hemocoel prior to host death [13]. Pinpointing the precise timing of gut leakage relative to host death is rendered difficult by the fact that the aforementioned observations were performed by invasive techniques which killed the insect before its death following *B. thuringiensis* exposure, or on host insects which had already died of it. 

A non-invasive assay for the study of gut leakiness relying on the use of a food dye that does not cross the healthy intestinal barrier was developed in *Drosophila melanogaster* by Rera et al. [14]. It has been called the “Smurf assay”, after the characteristic blue look exhibited by flies whose gut contents leaked into the hemocoel due to loss of intestinal integrity, which is a phenotypic marker for senescence in this species [14,15]. This assay was later adapted to two other fly species as well as *C. elegans* and *Danio rerio* [16]. It has also been used to successfully reveal the loss of gut wall integrity caused by a *Pseudomonas entomophila* oral exposure in larvae of *D. melanogaster* [17]. The adaptation of the Smurf assay in the context of a *B. thuringiensis* exposure could potentially reveal whether there is variability in the host in the susceptibility to gut leakage, as well as its timing relative to host death.

This is what we address in the present study: We investigated whether the Smurf assay could be adapted to two species of Tenebrionid beetles, the mealworm beetle *Tenebrio molitor* and the red flour beetle *Tribolium castaneum*, during an oral exposure to *B. thuringiensis*. We did so on hosts which are expected to vary in their susceptibility to *B. thuringiensis*-induced gut leakage. In *T. molitor*, the Cry3Aa toxin produced by *B. thuringiensis* var. *tenebrionis* [18] binds receptors on the midgut epithelium cells [19], leading to their lysis [20]. A decrease in *B. thuringiensis*-induced death in later larval stages has been shown in several insect species, probably due to a different amount of toxin binding sites on the midgut epithelium cells, or different toxin degradation capacities [21,22,23]. We therefore, first compare the mortality and occurrence of Smurfs in young versus older *T. molitor* larvae during *B. thuringiensis tenebrionis* exposure. Second, *T. castaneum* larvae have been shown to die less when exposed to a spores and crystals mixture of *B. thuringiensis* var. *tenebrionis* than of *B. thuringiensis* var. *tolworthi*. This might due to a higher affinity of the receptors of the midgut epithelial cells of *T. castaneum* for Cry3Ba, expressed by the var. *tolworthi*, than for Cry3Aa of *tenebrionis* [24,25]. We asked whether this was the case in our system, and whether this translated into a higher propensity for *B. thuringiensis* var. *tolworthi* to cause gut leakage compared to *B. thuringiensis* var. *tenebrionis*. 

## 2. Materials and Methods

### 2.1. Insect Rearing

*Tenebrio molitor* insects were purchased from Vivara Ltd (Vierlingsbeek, The Netherlands). Larvae were kept as an outbred stock at 25 °C in the dark and 70% relative humidity on wheat bran with weekly addition of albumin from chicken egg white as a protein source (Sigma-Aldrich, St Louis, MO, USA, ref A5503), organic white wheat flour and dried brewer’s yeast and a piece of apple at a density of 500 larvae per 2 L of wheat bran. Pupae were regularly retrieved from the stock and kept separately until completion of metamorphosis. Resulting adults were maintained individually in compartmented Petri dishes (Sterilin™, Thermo Fisher Scientific, Waltham, MA, USA) for 10 days until maturity before being regrouped for reproduction in our maintenance conditions. The larvae used in this study are the first generation born in the laboratory. *Tribolium castaneum* insects were collected from a flour mill in Croatia in 2010 (“CRO1” population, [26]) and maintained in the laboratory in organic white wheat flour complemented with 5% brewer’s yeast as a substrate and feeding source, at approximately 30 °C in a 12 h light–dark cycle. The larvae used in this study originate from egg laying events lasting for 6 h realized by a subsample of this population. Both our host species were used at an age where they will not reach pupation over the course of the experiment (i.e., are still at an active stage of growth), which prevents possible differences in toxicity between stages to be due to cessation of feeding [21].

During exposure to chemicals and bioassays, the 48-well plates, in which larvae were kept, were maintained in an opaque black plastic box allowing for air exchange and placed in an incubator providing the experimental conditions of the maintenance of our stock population.

### 2.2. Establishment of a Positive Control for Gut Disruption in T. molitor and T. castaneum

In order to determine whether gut leakage could be monitored by a Smurf assay in our two host species, we chemically-induced gut disintegration in 10 larvae of *T. molitor* whose size ranged from 7 to 8 mm, and 10 larvae of *T. castaneum* which were 20 days old. The choice of these stages was done to represent the sizes of larvae towards the end of the time course of our main experiments, ensuring that the migration of blue dye would be detectable in bigger larvae.

Calcofluor white has been shown to disrupt the peritrophic matrix by binding chitin in several Lepidoptera species [27], whereas sodium dodecyl sulfate (SDS) is known for its ability to damage the midgut epithelium of insects [28]. We; therefore, fed larvae with a synthetic exposure diet composed of 0.15 g agarose melted in 5 mL water, and then added with 4 mL of 20% SDS (Sigma-Aldrich, St Louis, MO, USA, ref. 74255), 4 mL of 0.1% Calcofluor white solution (Sigma-Aldrich, St Louis, MO, USA, ref. 18909), 0.23 g of blue dye (FD&C blue dye #1, Sigma-Aldrich, St Louis, MO, USA, ref. 861146) and topped up to 15 mL with 2 mL phosphate-buffered saline (PBS). The concentration of blue dye was chosen based on preliminary experiments in which we could easily observe the coloration through the cuticle of the larvae when in the digestive tract, as well as up to at least a 15-fold dilution injected into the hemolymph (data not shown). SDS and Calcofluor white solutions were replaced with 8 mL water in the corresponding control diet, which was also fed to 10 larvae of each host species.

Small cubes (0.3 × 0.3 × 0.3 cm) were cut from the solidified medium described above and fed to both *T. molitor* and *T. castaneum* larvae.

### 2.3. Bioassays

#### 2.3.1. Insects

*T. molitor* has a highly plastic larval life history, the number of molts until metamorphosis ranging from 11 to 22, making the relationship between larval size and age difficult to establish [29]. We thus, decided to select larvae based on their size. We investigated the effects of *B. thuringiensis* var. *tenebrionis* exposure in 2 size classes of *T. molitor* larvae retrieved from a box where adults were left to reproduce for at least 3 weeks. We selected “small” larvae on the basis of their length ranging between 4 and 5 mm, whereas “large” larvae were between 6 and 7 mm long (checked on a random sample of 10 larvae per 48 larvae replicate). These larvae are approximately 2 weeks old.

By contrast, the variance in size at a given age in our population of *T. castaneum* was lower, making it relevant to control for age after the egg laying event described above. Larvae were retrieved by sieving them from the substrate at 15 days after egg lay. A preliminary experiment showed us it was not possible to realize this assay on younger larvae (13 days old) since mortality due to handling was too high at this stage (see Appendix A).

#### 2.3.2. Bacterial Cultures

Both *Bacillus thuringiensis* var. *tenebrionis* (strain 4AA1), similar to *B. thuringiensis san diego*, and *Bacillus thuringiensis* var. *tolworthi* (BGSC number: strain 4L1; NRRL number: HD-125, Burgess number HDB-8) were obtained from the Bacillus Genetic Stock Center, and kept as an aliquoted stock in 25% glycerol at −80 °C. For each replicate, an aliquot was taken out of the freezer, streaked on LB agar and incubated at 30 °C overnight. On the next day, 5 single colonies were inoculated into 3 mL of liquid *Bacillus* medium (15 g peptone, 2 g D-glucose monohydrate, 6.8 g KH_2_PO_4_, 8.7 g K_2_HPO_4_ for 1 L distillated water, pH = 7.2) and incubated overnight at 30 °C with 200 rpm agitation, alongside a non-inoculated control. This starter culture was used to inoculate 500 mL of *Bacillus* medium and 2.5 mL salts solution (1.23 g MgSO_4_.7H_2_O, 0.02 g MnSO_4_.7H_2_O, 0.14 g ZnSO_4_.7H_2_O, 0.2 g FeSO_4_.7H_2_O in 50 mL ultrapure water) in Erlenmeyer flasks. After 3 days, 2.5 mL of salts solution were added again, alongside 625 µL 1 M CaCl_2_ solution.

*B. thuringiensis tenebrionis*, used to expose *T. molitor*, was grown at 25 °C, the temperature used for the maintenance of this host species, with a 200 rpm agitation for 7 days; time by which more than 95% of the bacteria were sporulated, which we confirmed by microscopic observation. Spores were recovered by centrifugation of our liquid culture at 4000× *g* for 10 min and washed in PBS 2 times. A serial dilution of the resulting solution was plated and incubated overnight, and the number of resulting colony forming units (CFUs) counted the next day. The concentration of spores was adjusted to 5 × 10^9^; CFU/mL before use.

*B. thuringiensis tenebrionis* and *B. thuringiensis tolworthi* strains, used to expose *T. castaneum*, were grown at 30 °C, with a 200 rpm agitation for 5 days until more than 95% of the bacteria were sporulated, recovered by centrifugation similarly to the protocol described above, and the final concentration of both strains also adjusted to 5 × 10⁹ CFU/mL before use; a concentration that elicited a high mortality in our insect hosts in preliminary experiments.

#### 2.3.3. Bacterial Exposure

Both host species were exposed to *B. thuringiensis* by being fed ad libitum agarose cubes containing blue dye and embedded spores-crystals mix, and maintained individually in the wells of a 48-well plate (Sarstedt, ref. 83.3923.005). This diet was unlikely to contain enough nutrients to allow the insects to complete their life cycle [29], but still supported an average of 2 molts per larva in both species. Our main concern was to maximize mortality upon ingestion of spores and crystals of *B. thuringiensis*, which has been shown to be impaired by the presence of particles in the exposure diet [30]. The cube was changed every second day, or as soon as a larva consumed a full cube. We think this protocol prevents that the insects avoid patches of contaminated food, which would bias mortality [31]. Larvae of both *T. molitor* and *T. castaneum* stop feeding shortly before molting and resume feeding shortly after, which was easily seen by absence of blue food bolus in the digestive tract [32]. Since some started a molt on the first day of the bioassay, there were larvae which had not fed on the first day. By the second day; however, all larvae had been feeding on the agarose cube. To embed the spores-crystals mix in the agarose, we boiled 0.2 g of agarose in 10 mL distilled water and added 0.3 g of blue dye. When this mix reached a temperature of approximately 50 °C, 5 mL of it were mixed with the spores-crystals mix, whereas the other 5 mL were mixed with 5 mL of PBS (control diet). This resulted in 10 mL of agarose with 2.5 × 10^9^; spores/mL and 10 mL of control diet. Cubes of this agarose were cut and fed to the larvae in the well of the plates where they were maintained. The larvae were observed daily for 7 days under a stereo microscope (Olympus SZX12, Tokyo, Japan), and pictures were taken with an Olympus SC50 camera when relevant. As stated by Martins et al. [33], the Smurf phenotype is a continuous phenotype, meaning that the blue shade in the hemolymph can be more or less pronounced. We scored as Smurfs the larvae that showed any blue hue in the hemolymph, observed under a binocular with transillumination, by opposition to a well-delimited blue gut in healthy individuals. This method avoids mistaking blue-stained cuticle with internal blue coloration; however, larvae that appeared blue were cleaned in water and re-observed before being scored as Smurfs or non-Smurfs. One limit of the method is that as time passes after death, it becomes increasingly difficult to identify the presence of blue dye in the hemolymph due to the melanization of the larva. In both our host species, we could reliably identify Smurfs around the time of death within 24 h, but they became hard to identify 48 h later (see Appendix A).

For each treatment (control and *B. thuringiensis tenebrionis* in the case of *T. molitor*; control, *B. thuringiensis tenebrionis* and *B. thuringiensis tolworthi* in the case of *T. castaneum*) larvae were maintained in 48-well plates, and this process replicated twice. Each replicate is composed of larvae originating from egg laying events that were separated in time, and exposed to bacterial cultures originating from 2 different aliquots of the bacterial stock. A few larvae got lost before our first sampling point resulting, in *T. molitor*, in a final sample size of 96 *B. thuringiensis tenebrionis* exposed and 94 control 4–5 mm larvae; and 96 *B. thuringiensis tenebrionis* exposed and 92 6–7 mm larvae. In *T. castaneum*, there were in total 96 *B. thuringiensis tenebrionis* exposed, 96 *B. thuringiensis tolworthi* exposed and 96 control 13-day-old larvae; and 96 *B. thuringiensis tenebrionis* exposed, 96 *B. thuringiensis tolworthi* exposed, and 92 control 15-day-old larvae.

### 2.4. Statistics

All statistical analyses and graphical representations were performed using the R software [34]. The survival of *T. molitor* larvae of both age/size classes was analyzed with a Cox mixed model (package “coxme”, [35]) for proportional hazards. In the case of *T. castaneum*, the hazards were not proportional between exposure treatments. We fitted an accelerated failure time model for a Weibull distribution with the “survreg” function of the “survival” package [36,37]. In both host species, the time and occurrence of death were included as response variables, according to the size/age class of the larvae and the exposure treatment they received as explanatory variables. As we sought to put in relation the timing of observation of gut leakage with the death of the larvae, we used the same approach to analyze the occurrence of Smurf phenotypes [15] along time with a Cox model for proportional hazards in both *T. molitor* and *T. castaneum*, according to the same explanatory variables and random factors described above. When the larvae died without having shown a Smurf phenotype, they were censored at the time of death.

Model selection was achieved by comparing the Akaike’s information criterion (AIC) of the full models including interactions to all the nested models and the null model. We kept as the best models the ones with the lowest AICs [38]. The Kaplan–Meier curves were made using the “ggplot2” package [39] and the “survminer” package [40].

## 3. Results

### 3.1. Chemical Induction of a Smurf Phenotype in T. castaneum and T. molitor

The treatment of both host species with SDS and Calcofluor lead to a distinctive Smurf phenotype, in which larvae showed a diffuse blue coloration of the body cavity, unlike control larvae which exhibited a well-delimited blue digestive tract in both *T. molitor* (Figure 1) and *T. castaneum* (Figure 2). The onset of gut disintegration happened earlier in *T. castaneum* than in *T. molitor*. We observed the first larvae exhibiting the characteristic Smurf phenotype 12 h after the beginning of the experiment in *T. molitor*, whereas we could already observe Smurfs as early as 6 h in *T. castaneum* larvae. More than 60% of the larvae showed the Smurf phenotype 24 h after exposure. We concluded that the leakage of the digestive tract content into the hemolymph was detectable by a Smurf assay in both beetle species. 

### 3.2. B. thuringiensis tenebrionis Exposure in Two Age/Size Classes of T. molitor Larvae

The effect of *B. thuringiensis tenebrionis* exposure over seven days differed between age/size classes (treatment × size class: X^2^_5,372_ = 6.14; *p* = 0.013; Figure 3). First, we can notice that, in the control treatments, mortality was higher in small than in bigger larvae (19% vs. 1%), indicating that, as expected, juvenile stages suffer a higher mortality during their development. Large larvae (6–7 mm long) took a longer time to die than small larvae, since it took the full seven days’ time course of the experiment for 50% of the large larvae to die versus approximately half that time for small larvae. Large larvae suffered a lower mortality during bacterial exposure than small larvae, as there was a 59% lower survival in large larvae exposed to *B. thuringiensis tenebrionis* compared to control versus a 74% lower survival in small larvae.

We found an effect of the exposure of *T. molitor* larvae to *B. thuringiensis tenebrionis* (treatment: X^2^_2,374_ = 8.92; *p* = 0.003; Figure 3) on the occurrence of Smurf phenotypes. However, it was very rare in our setting (less tan 10% of the larvae of both size classes). Thus, it appears that most of the dead larvae were not the ones whose dye had migrated in their hemocoel. We found no evidence for an effect of the size of *T. molitor* larvae either in interaction with the exposure treatment (treatment × size class: X^2^_5,372_ = 0.017; *p* = 0.9), or as a simple effect (size class: X^2^_2,374_ = 0.2; *p* = 0.65), indicating that, despite a higher mortality, the Smurf assay does not show a higher occurrence of gut leakage in small larvae.

In the few larvae showing a Smurf phenotype, we only observed the presence of blue dye in the body cavity simultaneously with the death of the larvae. Considering our sampling protocol, this could mean that the few larvae that presented the Smurf phenotype died of gut disintegration, or that it happened shortly after death. This phenotype was more or less pronounced between larvae, which probably reflects the quantity ingested by the larvae before death. By contrast, the phenotype we observed in the majority of the cases in moribund larvae and after death was a darkening of the midgut region resembling melanization, sometimes being expelled with the feces. Figure 4 illustrates these observations.

### 3.3. B. thuringiensis tenebrionis and B. thuringiensis tolworthi Exposure of T. castaneum Larvae

There was a significant effect of the bacterial exposure treatment on the mortality (treatment: X^2^_3,280_ = 49.9; *p* < 0.01). The mean survival time of larvae exposed to either *B. thuringiensis tenebrionis* or *B. thuringiensis tolworthi* was lower compared to control (z = −5.60; *p* < 0.001 and = −2.75; *p* = 0.006 respectively). Mortality dynamics differed between these two treatments, with *tolworthi*-exposed larvae living significantly longer than *tenebrionis*-exposed larvae (z = 4.05; *p* > 0.001). Indeed, larvae exposed to *tenebrionis* showed a progressive increase in percent mortality with time, similarly to what was observed in *T. molitor*, taking between five and six days for 50% of the larvae to die (Figure 5) and a mortality of 72% at the end of the seven days. By contrast, larvae exposed to *tolworthi* reached only a 43% mortality after seven days, with significant mortality events occurring only starting from day six of exposure. Thus, it seems that the peak mortality was happening at the time when we stopped following the experimental individuals, indicating that *B. thuringiensis tolworthi* causes a delayed but steeper mortality than *B. thuringiensis tenebrionis* (Figure 5).

Similarly to what we observed in exposed *T. molitor* larvae, the occurrence of the Smurf phenotype did not exceed 10% for both bacterial treatments. It was not affected by the exposure treatment (treatment: X^2^_3,280_ = 2.82; *p* = 0.25). The most often observed phenotype was a darkened digestive tract, which was present in live moribund larvae as well as in dead ones (Figure 6).

## 4. Discussion

With this series of experiments, we aimed at establishing the Smurf assay as a way to monitor the pathology caused by a *B. thuringiensis* exposure in *T. molitor* and *T. castaneum*. Surprisingly, the mortality caused by our exposure treatments was not mirrored by a high occurrence of Smurf phenotypes. This might have several causes. 

First, minor gut leakage might have gone undetected by the simple Smurf assay. However, positive controls based on chemically-induced gut leakage (cf. Materials and Methods) showed that migration of the blue dye into the hemolymph of our host species was detectable by the Smurf assay. Moreover, a few individuals showing a Smurf phenotype upon infection could be identified with our method, indicating that the assay generally worked. Likewise, in *D. melanogaster* larvae, the Smurf assay could successfully reveal the loss of gut wall integrity induced by an exposure to *P. entomophila* [17]. While chemically-induced gut leakage in our positive control is likely different from the ones induced by *B. thuringiensis* virulence factors, the inhibition of stem cell proliferation for gut epithelium renewal, and thus gut repair, is likely to be responsible for its leakiness during a *P. entomophila* oral exposure [41]. This repair mechanism is also responsible for resistance to *B. thuringiensis* in Lepidoptera [42], indicating that the damage caused by both these pathogens might share common mechanisms, in which case we could expect them to be both be detectable by a Smurf assay. 

Alternatively, the concentration of the Cry toxin relative to the concentration of spores might have been too low to cause gut leakage in most of our host larvae. This is unlikely considering the fact that our endpoint mortality is similar to the mortality achieved in other studies on *T. molitor* by exposure to a high concentration of the toxin alone [31,43,44], whereas the mortality caused by spores alone has been shown to be relatively low in other insect species [45].

It is therefore, possible that the low occurrence of Smurf phenotypes reveals that extensive gut disintegration, allowing the massive migration of gut contents containing bacteria into the hemolymph of the larvae, is not a widespread killing mechanism of *B. thuringiensis* var. *tenebrionis* and var. *tolworthi* in our host insects. If this is the case, our experiments could add further support to the observations made by [46], who pointed out that evidence of proliferation of *B. thuringiensis* in the hemolymph of dead insect hosts is more often found than evidence of it in live ones, suggesting that the death-by-septicemia model might not be generalized to all host species. Instead, *B. thuringiensis* might act locally in the midgut, which could be the cause of the darkened gut we observed in most moribund and dead larvae. Instead, *B. thuringiensis* might act locally in the midgut, which could be the cause of the darkened gut we observe in most moribund and dead larvae. In this context, the vegetative cells face the gut defense system instead of the hemolymph defense system of the host. They could face selection pressures comprising an acidic lumen in the anterior midgut, prophenoloxidases of the gut, continuous production of reactive oxygen species, as well as AMPs of the gut epithelium and lysozymes [47,48,49,50,51]. This does not exclude that the crossing of a few bacterial cells from the gut to the hemolymph could cause septicemia, being sufficient to cause host death with marginal gut leakage. An experimental setup involving the tracking of individual bacterial cells, using for example fluorescent bacteria, could give more insight on this aspect of the host pathogen interaction.

Consistently with what was observed in previous studies in Lepidoptera species [22,23,52,53], the exposure to *B. thuringiensis tenebrionis* caused more mortality in younger/smaller larvae of *T. molitor* than in larger/older ones compared to their respective control.

Stage dependent mortality to *B. thuringiensis* could be attributed to a reduced expression of binding sites on the midgut epithelium [22], or to an increase in binding site concentration with age instead [54]. Additionally, midgut extracts from late instars of some Lepidoptera degrade the toxin more efficiently [23,55]. More functional studies would be required to assess whether similar mechanisms explain the reduced susceptibility of larger/older *T. molitor* larvae to *B. thuringiensis* var. *tenebrionis*, as well as to determine with certainty whether this effect is mediated by larval size or larval instar. 

Contrary to a previous study [24], *T. castaneum* larvae exposed to *B. thuringiensis tenebrionis* showed a higher mortality after seven days than *B. thuringiensis tolworthi* exposed ones. In Contreras et al. [24], the *T. castaneum* larvae were of approximately the same age as in our study, but were of a different population (Ga-2), and were exposed to spores and crystals via flour discs. However, another study by Milutinovic et al. [26], using flour discs as a mode of exposure, found a result similar to ours, where in similar spore concentration range *tenebrionis* caused more mortality to *T. castaneum* than *tolworthi*, which yielded almost no mortality. In the latter study, the authors used a different isolate of *B. thuringiensis* var. *tolworthi*, whereas the *T. castaneum* larvae originated from the same population as in our study. This indicates, as expected, that isolates of *B. thuringiensis* var. *tolworthi* differ in their virulence towards *T. castaneum*, and that, as previously stated in Lepidoptera species [21], the virulence of serovars of *B. thuringiensis* differs between our host insect populations. 

More interestingly, in our case, the killing dynamics of the two bacterial strains differed, with *B. thuringiensis tenebrionis* causing a constant mortality over time, whereas larvae exposed to *B. thuringiensis tolworthi* showed less variability in the time to death, causing a more dramatic mortality peak at the end of the time course. The addition in the exposure diet of the Cry3Aa-binding fragment of *T. molitor*’s cadherin receptor increases the toxicity of the Cry3Aa toxin in *T. castaneum* larvae [56], showing that the binding affinity of the toxin to its receptors is primordial for *B. thuringiensis tenebrionis* induced mortality. This suggests first that other virulence factors of the bacterial cells of *B. thuringiensis tenebrionis* compensate for the lower binding affinity of the Cry3Aa delta-endotoxin with the midgut epithelium cells receptors of *T. castaneum* [25]. This also points out to a different mode of action of the virulence factors of *B. thuringiensis tolworthi*, and/or different dynamics of the bacterial population in the host. A promising lead would be to investigate whether *B. thuringiensis tolworthi* would reach a higher fitness than *B. thuringiensis tenebrionis* in *T. castaneum*, for example, in terms of number of spores produced or spore viability, as a result of more efficient virulence factors.

One other possible reason for host death can be the impairment of larvae development by the proliferating *B. thuringiensis* [57], which can also cause the cessation of feeding [58]. This effect might be accentuated by the poor nutritional source used in this assay. In a previous study on *D. melanogaster* [17], larvae from selection lines showing a higher resistance to starvation also show a higher mortality to the food-borne pathogen *P. entomophila*, mirrored by a higher occurrence of gut leakage in the host larvae. It is therefore possible that in our system, the poor nutritional value of the food source increases the proportion of Smurf phenotypesmeaning their occurrence might even be lower during a bioassay using a proper food source as a vector for exposure.

## 5. Conclusions

Our study reveals a higher mortality caused by *B. thuringiensis* var. *tenebrionis* in young *T. molitor* larvae compared to older ones. It also highlights a different killing dynamics of *B. thuringiensis* var. *tolworthi* compared to *B. thuringiensis* var. *tenebrionis* in *T. castaneum* larvae. However, the adaptation of a Smurf assay to a *B. thuringiensis*/tenebrionid beetles system did not reveal gut leakage in host larvae during bacterial exposure. This could mean that this assay is not adapted to the system, or alternatively, that extensive gut leakage is not a widespread killing mechanism of *B. thuringiensis* in *T. molitor* and *T. castaneum*. More functional tests would be needed to determine whether death occurs through septicemia, toxemia, or starvation due to arrested feeding. This could help identifying the selection pressures faced by the bacterium during its proliferation in the host.

## Figures and Tables

**Figure 1 insects-11-00261-f001:**
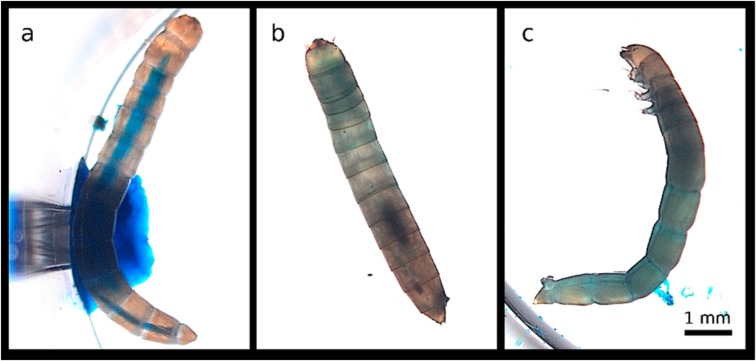
Range of phenotypes exhibited by *T. molitor* larvae upon ingestion of the positive control diet. (**a**) Healthy control *T. molitor* larva with a well-defined digestive tract filled with blue-dyed agarose (**b**) Dead larva showing a subtle Smurf phenotype. (**c**) Dead larva showing a strong Smurf phenotype upon ingestion of the positive control diet. The larvae chosen for this experiment were 7–8 mm long and died within one day of exposure. Larva on panel (**b**) retracted after death and appears smaller than its initial size.

**Figure 2 insects-11-00261-f002:**
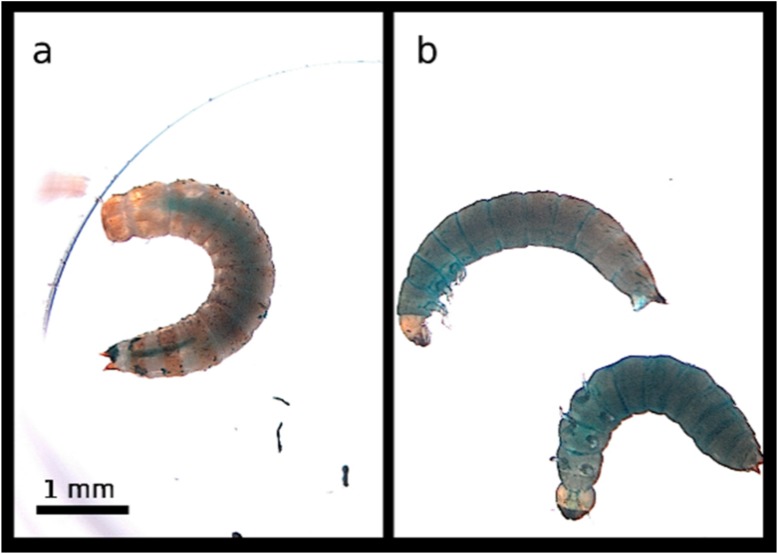
Range of phenotypes exhibited by *T. castaneum* larvae upon ingestion of the positive control diet. (**a**) Healthy control *T. castaneum* larva with a well-defined digestive tract filled with blue-dyed agarose. (**b**) Dead larvae showing the Smurf phenotype upon ingestion of the positive control diet. The larvae were 20 days old at the beginning of the experiment and died within one day of exposure.

**Figure 3 insects-11-00261-f003:**
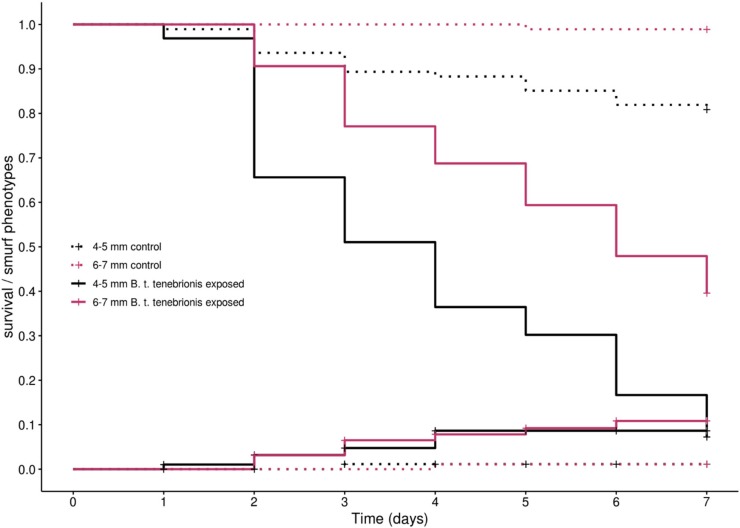
Survival and occurrence of Smurf phenotype in small and large larvae of *T. molitor* during exposure to *B. thuringiensis tenebrionis*. Kaplan-Meier curves representing the survival (descending curves) and the cumulative incidence of Smurf phenotypes (ascending curves) in *T. molitor* larvae of 4–5 mm (in pink) or larvae of 6–7 mm (in black) when exposed to a control diet (dashed lines) or 2.5 × 10^9^ CFU/mL of *B. thuringiensis* var. *tenebrionis* spores and crystals mix (plain lines). Data in Appendix A.

**Figure 4 insects-11-00261-f004:**
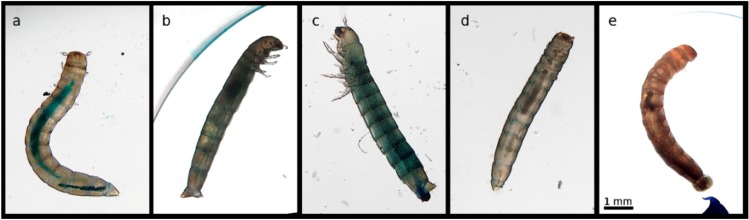
Phenotypes of *T. molitor* larvae observed under the experimental conditions. (**a**) Healthy control larva with a well-defined digestive tract filled with agarose and blue dye. Unlike (**b**), dead larva, showing a subtle Smurf phenotype upon exposure to *B. thuringiensis.* (**c**) Dead larva showing a stronger Smurf phenotype upon bacterial exposure. (**d**) Dead larva showing a darkened digestive tract upon *B. thuringiensis* exposure; the phenotype we observed the most frequently. (**e**) Moribund larva with expelled darkened gut content. The pictures represent larvae belonging to different size classes (**b**,**d**) 4–5 mm; (**a**,**c**,**e**) 6–7 mm); however, they are representative of the range of phenotypes present in both size classes. Their size on the picture differs from their size at the beginning of the experiment, since they kept growing before dying at different times after exposure. The scale represented on the rightmost picture is common to all the panels of the figure.

**Figure 5 insects-11-00261-f005:**
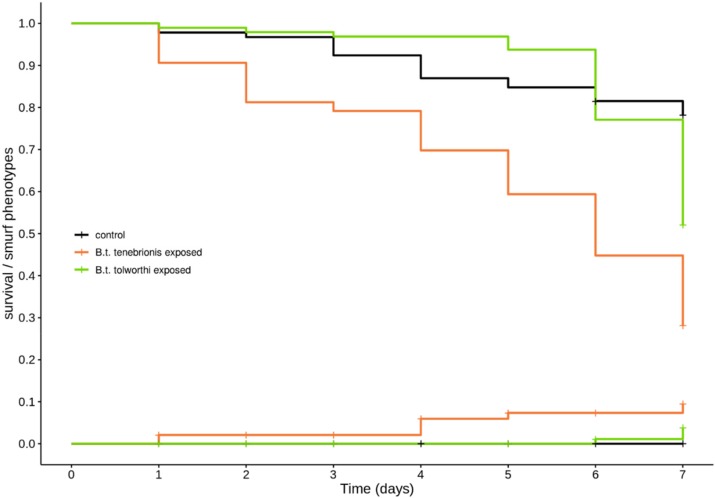
Survival and occurrence of Smurf phenotype in 15-day-old *T. castaneum* larvae exposed to *Bacillus thuringiensis* var. *tolworthi* and *B. thuringiensis* var. *tenebrionis*. Kaplan–Meier curves representing the survival (descending lines) and the cumulative incidence of Smurf phenotypes (ascending lines) in 15-days-old *T. castaneum* larvae exposed to a control diet (in black), 2.5 × 10^9^ CFU/mL of *B. thuringiensis* var. *tenebrionis* (in orange) and 2.5 × 10^9^ CFU/mL of *B. thuringiensis* var. *tolworthi* (in green) spores and crystals mix. Data in Appendix A.

**Figure 6 insects-11-00261-f006:**
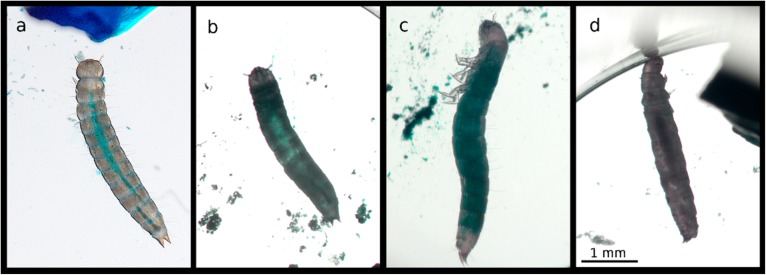
Phenotypes of *T. castaneum* larvae observed under the experimental conditions. (**a**) Healthy control larva with a well-defined digestive tract filled with blue-dyed agarose. (**b**) Dead larva showing a subtle Smurf phenotype upon *B. thuringiensis* (here *tenebrionis*) exposure. (**c**) Dead larva showing a stronger Smurf phenotype upon *B. thuringiensis* (here *tenebrionis*) exposure. (**d**) Dead larva showing a darkened digestive tract upon *B. thuringiensis* (here *tolworthi*) exposure; this is the phenotype we encountered the most frequently. The pictures represent larvae exposed to either *B. thuringiensis tenebrionis* or *tolworthi*; however, they are representative of the phenotypes induced by both *B. thuringiensis tenebrionis* and *tolworthi*. The scale represented on the rightmost picture is common to all the panels of the figure.

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
