# Peer review of "Comparative Mortality and Adaptation of a Smurf Assay in Two Species of Tenebrionid Beetles Exposed to Bacillus thuringiensis"

_insects, 2020, doi:10.3390/insects11040261_

Round 1

Reviewer 1 Report

The manuscript empirically tests if Bacillus thuringiensis induced in two species of tenebrionid beetles is due to loss of intestinal integrity, and attempts to determine if differences in host susceptibility (stage-specific, species-specific, and pathogen strain specific) when infected with the pathogen depends on the extent to which the intestine is damaged. For this, the authors adapt a dye-based assay from Drosophila studies called ‘smurf’ assay.

Through their assays the authors clearly show that differential mortality observed in most cases that they test is not correlated with loss of gut integrity. Given our experience with Drosophila I however think that if the host is nutritionally stressed the effect on gut might differ. But since this angle of investigation is beyond the scope of this study, I suggest the authors add a few sentences on dietary effects on pathogen induced smurf phenotype. Nevertheless, these results are very important, especially since it questions the common assumption that BT induced insect mortality is mainly due to gut disintegration. The result clearly challenges this view with empirical data suggesting that further studies are required to determine the underlying physiological cause.

I have no specific issues with this study. A few typos and repeated words in some sentences need to be edited. It would however be informative to the reader if the authors expand their statement in the conclusion “This does not exclude that the crossing 385 of a few bacterial cells from the gut to the hemolymph could cause septicemia, and more functional 386 tests would be needed to determine whether death occurs by septicemia, toxemia, starvation due to 387 arrested feeding [57] or impairment of larval development [58]. This could help identifying the 388 selection pressures faced by the bacterium during its proliferation in the host.” They can do this in the discussion.         

Author Response

We would like to thank the reviewer for taking the time to improve this manuscript. We believe we have corrected typos and repetitions.

Our statement in the conclusion is now expended in the discussion, lines 432 to 436.

We furthermore discuss the effect of nutritional restriction in the light of the paper cited in reference number 17, lines 481 to 491.

Kind regards,

Caroline Zanchi

Reviewer 2 Report

Lines 23. Key words must some be in italics

Line 52. Live hostS…………..  kill through septicemia.

Line 63 ……….(add words) ….  assay for the study of gut …..

The authors should use past tense when writing the methods section. (as well as other text where relevant). For example:

line 142> are should be were.

Line 143   & 146 & line 176: is>was.  Line 203: are>were

Line 232: First, we observed that in …. mortality WAS….

Line 243. Rephrase. Unclear

Line 249. Smurf must be capital letter. Check complete manuscript please.

Line 322 &344. Lepidoptera higher case – check complete manuscript. However, in line 452, it should not be higher case.

Line 333. Rephrase : our host insects…

Fig. 2. These will be very nice photos and very clear if they were more transparent.

Line 287. Fig caption. T. castaneum must be Higher case.

Lines 125-133. This is Methods section  by  these are already results that are presented here.

No clear indication was provided in Methods section (statistical section) on how LT50s were determined.

There were many errors in the reference list in terms of Italics of scientific names.

Author Response

Lines 23. Key words must some be in italics

Done

Line 52. Live hostS…………..  kill through septicemia.

Done

Line 63 ……….(add words) ….  assay for the study of gut …..

Done

The authors should use past tense when writing the methods section. (as well as other text where relevant). For example:

line 142> are should be were.

Done

Line 143   & 146 & line 176: is>was.  Line 203: are>were

Done

Line 232: First, we observed that in …. mortality WAS….

Done

Line 243. Rephrase. Unclear

We hope we made it clearer

Line 249. Smurf must be capital letter. Check complete manuscript please.

Done

Line 322 &344. Lepidoptera higher case – check complete manuscript. However, in line 452, it should not be higher case.

Done

Line 333. Rephrase : our host insects…

Done

Fig. 2. These will be very nice photos and very clear if they were more transparent.

Tenebrio being more "opaque" than Tribolium, equally nice pictures are difficult to obtain. We believe we improved them while still fitting into the guidelines of MDPI.

Line 287. Fig caption. T. castaneum must be Higher case.

Done

Lines 125-133. This is Methods section  by  these are already results that are presented here.

We now present this as the first part of the results section.

No clear indication was provided in Methods section (statistical section) on how LT50s were determined.

What we wrote was indeed misleading. We did not perform a probit analyse, which would have been redundant with our main analysis. We simply looked at when 50 % of the larvae were dead. We rephrased this passage to make this clear.

There were many errors in the reference list in terms of Italics of scientific names.

We believe we corrected them all.

Thank you again on behalf of the authors,

Kind regards,

Caroline Zanchi